# Artificial Neural Network Controller for a Modular Robot Using a Software Defined Radio Communication System

**Luis Fernando Pedraza** [1,*], **Henry Alberto Hernández** [2] **and Cesar Augusto Hernández** [3]

1 Telecommunications Engineering Department, Universidad Distrital Francisco José de Caldas, Bogotá 11021-110231588, Colombia

2 Control and Automation Engineering Department, Universidad Distrital Francisco José de Caldas, Bogotá 11021-110231588, Colombia; hahernandezm@udistrital.edu.co

3 Electrical Engineering Department, Universidad Distrital Francisco José de Caldas, Bogotá 11021-110231588, Colombia; cahernandezs@udistrital.edu.co

* Correspondence: lfpedrazam@udistrital.edu.co

**Abstract:** Modular robots are flexible structures that offer versatility and configuration options for carrying out different types of movements; however, disconnection problems between the modules can lead to the loss of information, and, therefore, the proposed displacement objectives are not met. This work proposes the control of a chain-type modular robot using an artificial neural network (ANN) that enables the robot to go through different environments. The main contribution of this research is that it uses a software defined radio (SDR) system, where the Wi-Fi channel with the best signal-to-noise Ratio (SNR) is selected to send the information regarding the simulated movement parameters and obtained by the controller to the modular robot. This allows for faster communication with fewer errors. In case of a disconnection, these parameters are stored in the simulator, so they can be sent again, which increases the tolerance to communication failures. Additionally, the robot sends information about the average angular velocity, which is stored in the cloud. The errors in the ANN controller results, in terms of the traveled distance and time estimated by the simulator, are less than 6% of the real robot values.

**Keywords:** artificial neural network (ANN); modular robot; software defined radio (SDR); signal-to-noise ratio (SNR)

## 1. Introduction

Today, modularity is present in numerous areas of industry and robotics; therefore, modular systems offer benefits such as versatility, robustness and low-cost manufacturing compared to fixed-parameter conventional designs [1]. This has driven the use of modular robots whose structure is made up of multiple modules that are combined in different configurations to carry out various kinds of tasks. Some of these tasks include simple movements such as spinning or moving forward and complex movements such as walking or crawling [2,3]. The scope and movements of the robotic structure depend on the shape and number of degrees of freedom of each module, since these variables can increase the processing capacity required to synchronize the articulations of each module [4,5].

The techniques to control a modular robot can be centralized, decentralized or hybrid. In the first case, structure control is embedded in a single device [6]. In the second case, the controller can be any module in the structure [7,8]. In the third case, the controller incorporates features of the previous controllers; that is, there is a central controller that sends parameters to the modules that translate said information to perform a task [9–11].

The artificial neural network (ANN) has been used in robotics applications due to the high computation rate and capacity to support nonlinear functions. The uses of ANN controllers in robotics include the support of wireless connections for underwater swarm robots [12], the control of a five-degrees-of-freedom robot [13] and a neural-learning-based sensorless control scheme in the presence of an input dead zone for a robotic arm [14].

Furthermore, the evolution of communication systems throughout the years has led to their application in robots to enhance their performance [15]. One of the more novel approaches in communication systems corresponds to the development of software defined radio (SDR) technology, which is a radio system where the components are implemented using software to interact with hardware [16,17]. In this project, the Raspberry Pi 3 device was used to implement the SDR since it allows for the development of wireless applications with a robust low-cost embedded system, which has been used as a communication system for robot control in the monitoring and storage of data in real time [18]. It has also been used for the control and communication of a robot that produces basic motions and sends a video to an Android device [19]. Typically, communication in modular robots is based on infrared or wired communication; however, some initiatives have been developed to communicate the modules wirelessly using ZigBee [20] and Wi-Fi [21] technologies, although to date, there are no reports on the use of SDR communication.

This work contributes to the literature with the use of a wireless ANN controller that builds the path of an EMERGE modular robot in a simulator and sends the information to the modular robot through an SDR communication network implemented in a Raspberry Pi 3. The result of this is the approximation of the behavior of the robot before its start-up and in the use of a communication channel that causes fewer errors and has a higher speed than those around it, at a low cost. This paper is organized as follows: Section 2 describes the structure and operation of the system, while Section 3 presents and discusses the results obtained with the controller executed in the simulator and the robot. Lastly, Section 4 presents the conclusions derived from the overall work.

## 2. System Development

The elements that compose the system are described below: the EMERGE modular robot, the simulator, the ANN controller and the SDR communication platform, as shown in Figure 1. Furthermore, this section details the concepts required to understand the operation of the EMERGE modular robot, the control method and an explanation of the experiments carried out to evaluate the controller.

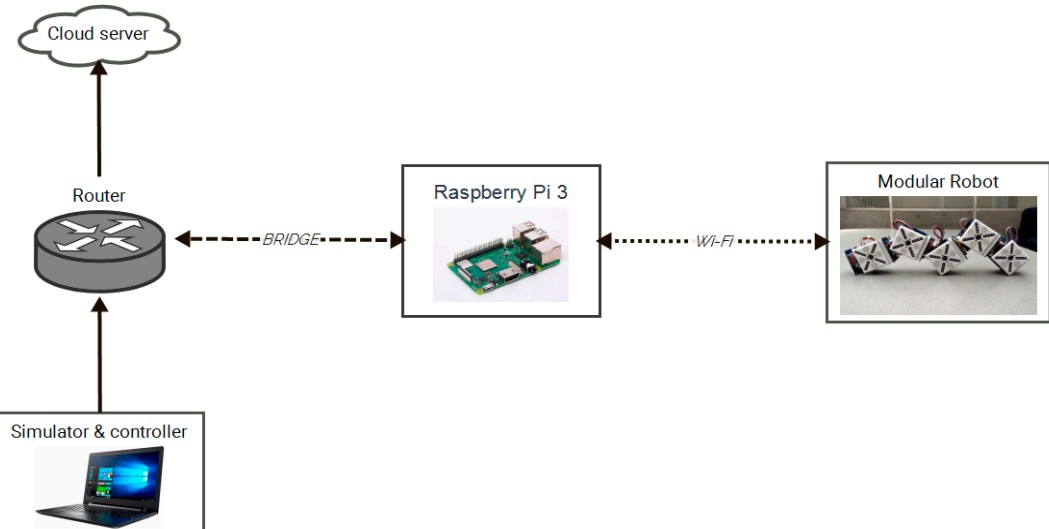

**Figure 1.** General diagram of the developed system.

## 2.1. EMERGE Modular Robot

The EMERGE robot seen in Figure 2a is an open-use prototype; that is, the materials, electrical circuits and procedure necessary for its assembly can be found in a repository [22]. Additionally, this prototype is flexible, which allows the user to adapt the circuits to particular needs [23–25]. For instance, in this case, a printed circuit was added to the robot with the ESP32 microcontroller, which allows it to communicate with the Raspberry Pi using Wi-Fi wireless technology.

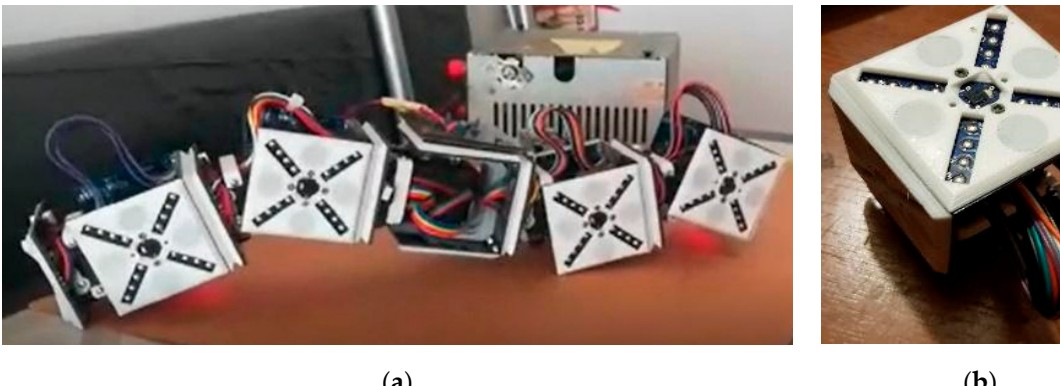

(**a**)          (**b**)

**Figure 2.** EMERGE modular robot: (**a**) Assembly; (**b**) Individual module.

This robot is basically composed of various modules, such as the one shown in Figure 2b, where the user defines the grouping. Each module has four sides with magnets that can be connected to other modules. The information is shared using the controller area network (CAN) protocol, which can package, send and receive the information from or towards a specific module or device [23–25].

Although, in this application, chain-type morphologies were considered to carry out the experiments, the CAN communication protocol and the structure of each module are flexible, which allows the robot modules to be grouped with different types of morphologies and, therefore, perform various tasks [22].

The traditional method to control the EMERGE robot is based on a centralized controller such as the one in Figure 3a, that is implemented in the platform, so the controller has an embedded control algorithm that is executed in real time during the operation of the platform [23]. However, when the platform is turned off or a module is disconnected, the controller is desynchronized and the robot stops moving. This is avoided with the control and communication strategy proposed in this paper.

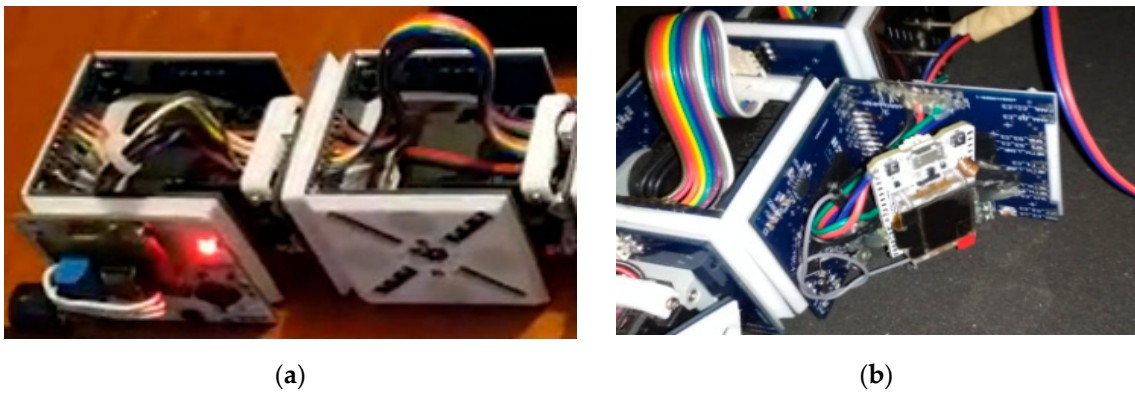

(**a**)          (**b**)

**Figure 3.** Local controllers: (**a**) Traditional; (**b**) Wi-Fi module.

The proposed controller modifies the centralized control technique using an evolutionary algorithm that is executed locally in the controller. This is achieved by generating the control parameters on

the computer, in which the parameters are simulated and sent to the robot through the SDR network, which connects to the robot's Wi-Fi module (ESP32 microcontroller), presented in Figure 3b. This device sends the received information to each module through the CAN bus [23].

## 2.2. Modular Robot Simulator

The simulation environment was developed in the 3D World Editor application in Matlab [26]. The dynamic interactions between the module chains, the environment and simulated obstacles are displayed in the editor. This allows the user to know in advance the real movements of the robot. Figure 4a shows the virtual module of the robot implemented in the 3D simulation. This module was created in the SolidWorks software. The virtual modules are coupled to create the robot morphology as shown in Figure 4b, and the movement is produced according to the rotation and translation data received from the controller.

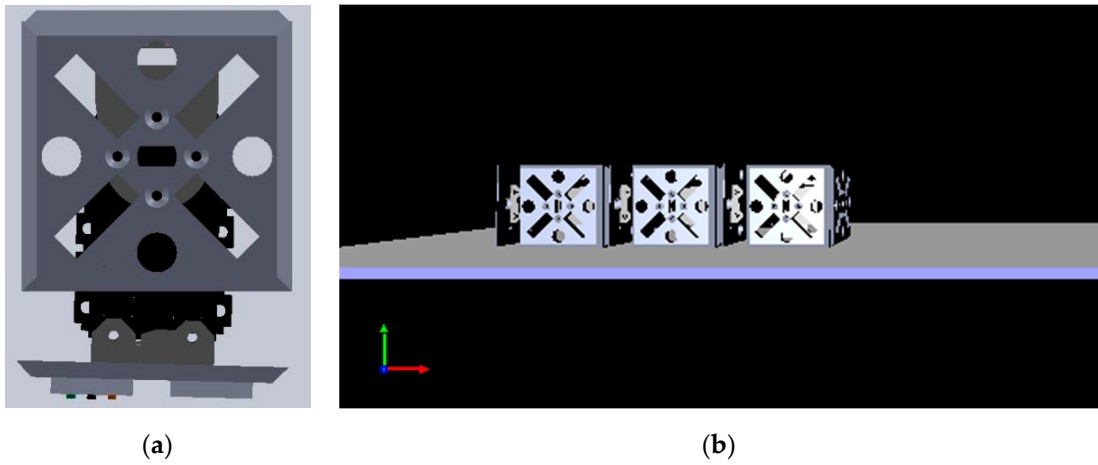

(**a**)                                                                 (**b**)

**Figure 4.** Virtual simulator: (**a**) Module developed in SolidWorks; (**b**) Design and assembly of the robot in the 3D World Editor environment.

The sequence of movements in the simulator is generated using the motion control tables, which were designed based on a sine function with an amplitude, frequency and phase shift for each module. As a result of this function, values between 0 and $\pi$ are obtained, and then a conversion is performed for the start-up of the actuator of the module, which receives values between 0 and 1024 as shown in Figure 5. The conversion is limited in the range of $[\frac{\pi}{4}, \frac{3\pi}{4}]$, since any value outside this range would compromise the mechanical structure of the robot.

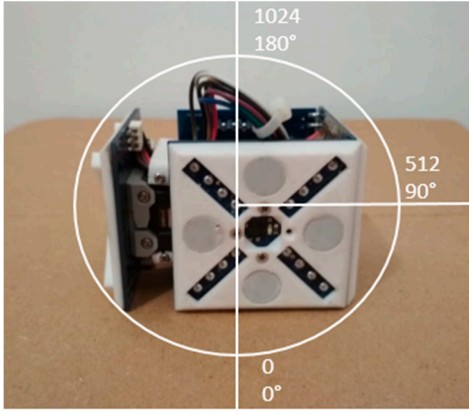

**Figure 5.** Motion range for the module actuator.

The simulator user interface shown in Figure 6 allows the selection of the morphology and environment to simulate the path. Three designed morphologies are available with 3 to 5 modules, as well as three environments: a flat surface, a ladder as an obstacle and an L-shaped path. Then, the ANN controller is executed to determine the position of each module of the robot. The data are represented graphically in the virtual environment, and, if needed, these are sent to the real robot through the SDR communication network.

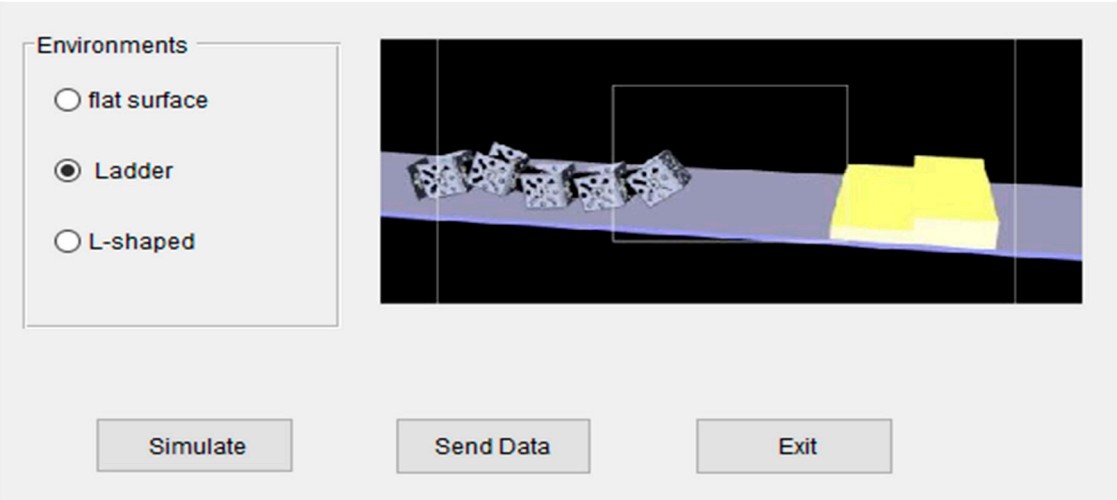

**Figure 6.** The simulator user interface.

### 2.3. Artificial Neural Network Controller

The ANN controller calculates the position of each robot module [27–30]. In this case, a backpropagation ANN was implemented as shown in Figure 7a, which uses a training algorithm based on the correction of the mean squared error. Basically, it is a margin of error ($\in$) that is estimated as the average of the squares of the errors, as shown in the following equations—that is, the difference between the expected value ($\hat{y}_i$), contained in a dataset, and the estimated value ($y_i$), calculated by the ANN. This difference is adjusted in each training iteration based on the weights ($w_i$) of each ANN neuron, with an activation function $f_i$, until the error is close to zero (in this case, a threshold of $\tau = 4 \times 10^{-8}$ is established).

$$y_i = f_i(w_i \times y_i + \epsilon) \tag{1}$$

$$\epsilon_i = \frac{1}{n} \sum_{i=1}^{n} (\hat{y}_i - y_i)^2 \tag{2}$$

In this case, the training dataset, presented in Table 1, is a database containing the following information: the number of modules, type of environment and position of the modules. Furthermore, this dataset contains 1000 records of samples taken during the operation of the robot modules in different environments, 750 records to be used for ANN training and 250 for validation of the estimated results. The ANN controller inputs are the type of environment and the number of modules, while the positions of the modules are the outputs. Afterward, the ANN is trained to reproduce the set of movements in the simulator and, if required, in the robot, as shown in Figure 7b. The number of hidden layers of the ANN was established as 25 since that is the minimum number to estimate a set of continuous and stable movements with a low error, as is analyzed in chapter 3, and using moderate computational resources during training. The training time is approximately 430 s.

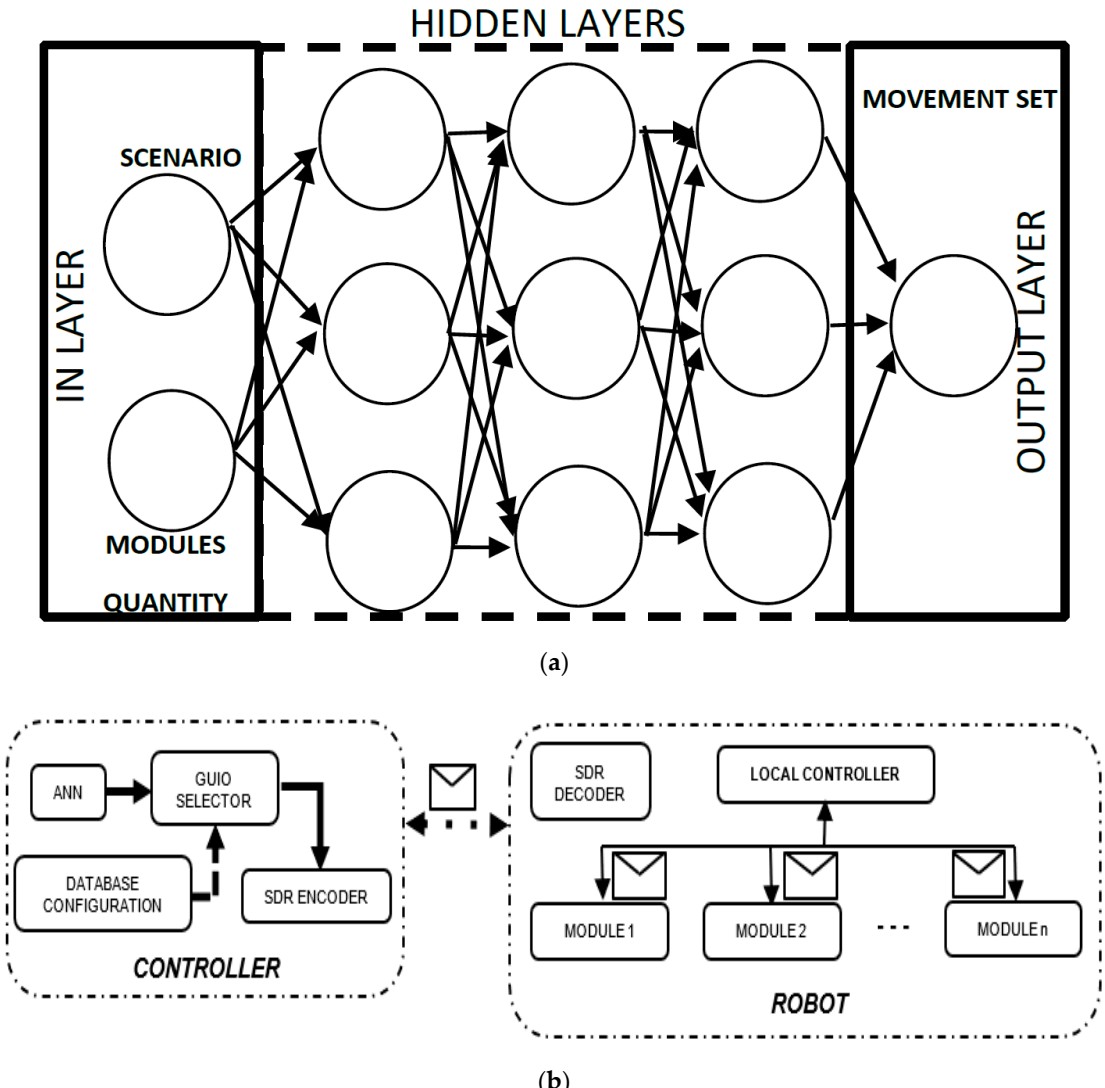

(**a**)

(**b**)

**Figure 7.** Artificial neural network (ANN) configuration: (**a**) Implementation of the layers; (**b**) Robot controller.

**Table 1.** Database segment used to train the ANN.

| Inputs | | Outputs | | | | |
|---|---|---|---|---|---|---|
| Scenario | Number of Modules | Module 1 | Module 2 | Module 3 | Module 4 | Module 5 |
| Flat surface | 3 | 300 | 320 | 340 | - | - |
| Flat surface | 4 | 325 | 345 | 365 | 385 | - |
| Ladder | 4 | 350 | 370 | 390 | 410 | - |
| L-shaped | 4 | 375 | 395 | 415 | 435 | - |
| Flat surface | 5 | 400 | 420 | 440 | 460 | 480 |
| Ladder | 5 | 425 | 445 | 465 | 485 | 505 |

In Algorithm 1, the instructions of the proposed ANN controller are presented, which delivers the movements to the simulator and, if required, to the robot. The three environments over which the robot can move, to evaluate the algorithm, are shown in Figure 8.

---

**Algorithm 1**. Control strategy

---

**Function** *ANN* (in *i*, out *o*, margin of error $\epsilon$)
    $\tau = 4 \times 10^{-8}$                                             //Threshold as stop condition
    *ANN* $\leftarrow$ *Inputs* $[i][1]$          //In vector layer
    *ANN* $\leftarrow$ *Outputs* $[1][o]$      //Out vector layer
    *ANN* $\leftarrow$ *Activation function* $(f_i)$   //$f_i$ is a gaussian function (Equation (1))
    *ANN* $\leftarrow$ *Hidden layers* $[10][25]$  //Matrix 10 neurons $\times$ 25 layers
    *ANN* $\leftarrow$ *Initial weights*        //Initial weight assignment function
    **While** $\tau < \epsilon$ **do**
        *ANN* $\leftarrow$ *optimize weights* $(i, o)$
        $\epsilon \leftarrow$ *test ANN*           //Validation of results
    **End While**
    **return** *ANN*
**End** *ANN*

**Function** Virtual enviroment ()
    Load libraries 3D enviroment          //Load virtual objects and robot
    Start SDR port                    //Open port to establish communication
    Create communication port read thread   //Start communication routine
    Create communication port send thread
    Start GUIO (Graphical User Interface Objects)  //Start program
    $i \leftarrow$ *number of modules*        //Select robot morphologie
    $j \leftarrow$ *select escenario*          //Select enviroment
    $m \leftarrow$ *mode*                 //Select routine test or *ANN* mode
    *epochs* $\leftarrow 0$                //Start iterations
    **If** *mode* $== 1$ **then**
        $\theta_i \leftarrow$ *movements database*   //Read predefined movements (Table 1)
    **else**
        $\theta_i \leftarrow ANN(i, j, inf)$     //Read *ANN* movements
                                   **//Note:** *inf* is a initial value (can be > 10)
                                   //for $\epsilon$ and start *ANN* weights
    **While** *epochs* $< 200$ **do**
        $\propto_i \leftarrow$ Generate movements in the virtual reality environment $(\theta_i)$
                                   //Nomalize the $\theta_i$ value and fixes it
                                   //on the actuator scale
            Send via serial port $(\propto_i)$    //Send $\theta_i$ to each real module
            Run the move routine for 100 milliseconds  //Delay for the next movement
        *epochs* ++
        $\epsilon \leftarrow MSE(\theta_i)$          //Mean Squares Error routine (Equation (2))
        $\theta_i \leftarrow ANN(i, j, \epsilon)$     //Simulates the *ANN* and update weights
    **End While**
**End** Virtual enviroment

---

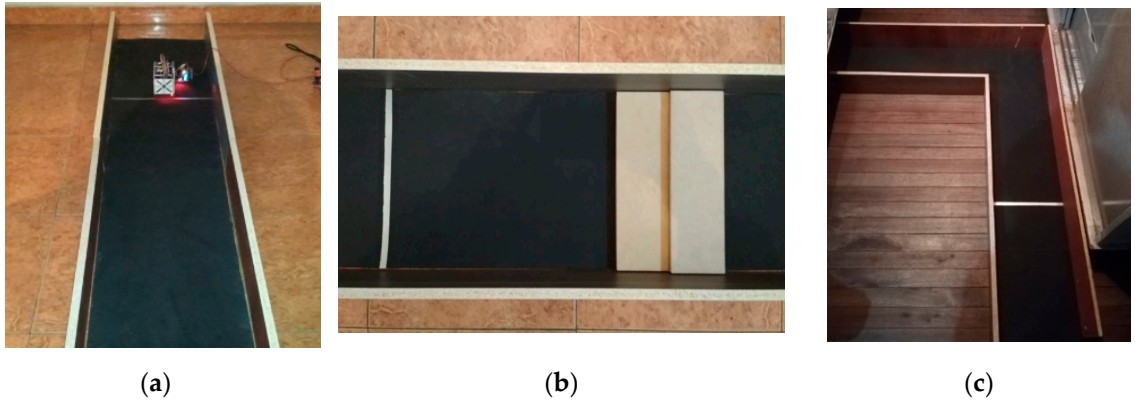

| (a) | (b) | (c) |

**Figure 8.** Environments used to evaluate the controller algorithm: (**a**) Flat surface; (**b**) Ladder as an obstacle; (**c**) L-shaped path.

## 2.4. Software-Defined Radio Communication System

The communications system developed, presented in Figure 1, is composed of the following. First, there is a Raspberry Pi 3, which is configured as a wireless access point (WAP) in which the SDR is performed and the wireless network is used to measure the power level of the channels of the Wi-Fi network. Second, there is a router that provides an internet connection and sends the information generated from the robot to the cloud. This router is also connected to a computer in which the ANN controller is simulated and developed to send movements to the robot. Third, there is an ESP32 microcontroller that communicates bidirectionally with the Raspberry, transmitting the motion sequence to the robot and the angular velocity to the Raspberry, to be sent and stored in the cloud.

In the WAP configuration of the Raspberry, the name of the wireless network (SSID), the channel, and the level of security, among other settings, are edited. To provide internet access through the WAP, a bridge between the Raspberry's wireless interface and the ethernet network adapter is created. Hence, the traffic is redirected through the network cable that is connected to the router to access to the Internet.

The Raspberry Pi has limitations in the network interface, so a USB dongle is used to measure the power of the surrounding wireless networks. After measuring the power of the channels in the 2.4 GHz band, this information is sent to the Raspberry Pi to start the SDR.

The SDR system establishes the Wi-Fi transmission between the Raspberry and the ESP32, from the beginning, using the channel with the highest SNR. To achieve this, the powers or received signal strength indicators (RSSI) are captured from the channels of the access points found in the Raspberry environment, which are measured by the USB dongle. Then, the power per channel is averaged based on the RSSI measurements of the access points, as shown below:

$$\overline{P}_{channel} = \frac{\sum_{i=1}^{N} P_{channel}}{N} \qquad (3)$$

where $\overline{P}_{channel}$ is the average power of the Wi-Fi channel in dBm, $P_{channel}$ is the channel power for a wireless access point in dBm and $N$ is the number of wireless access points that are present in the same channel. The Gaussian white noise power is now calculated [31]:

$$P_{white\_noise} = 10 \log (kTB) \qquad (4)$$

where $k$ is the Boltzmann constant $1.3806852 \times 10^{-23}$ J/K; $T$ is the ambient temperature in degrees Kelvin—in this case, it is 298.15 °K; and $B$ is the bandwidth of each Wi-Fi channel in Hz (20 MHz). Therefore, $P_{white\_noise}$ is −131 dBm. Finally, the SNR of each channel can be determined as [32]:

$$\text{SNR} = \overline{P}_{channel} - P_{white\_noise} \tag{5}$$

Then, the channel with the highest SNR is chosen and set to the access point.

The communication between the Raspberry and the ESP32 is bidirectional. The Raspberry sends the sequence of movements to the ESP32 located in the modular robot. The ESP32 sends the robot's average angular velocity to the Raspberry. The user datagram protocol (UDP) is used in this task.

The routine program in the Raspberry Pi was developed in Python, which directs the packets of the robot's motion sequence to the IP address of the ESP32, enables the input buffer to receive packets from the ESP32, and executes a sub-process to connect to the *ThingSpeak* servers and thus send the robot's performance parameters to the cloud, which, in this case, are the average angular velocity of each movement.

The programming algorithm contained in the ESP32 was developed in the Arduino IDE. This contains the necessary instructions to interpret the commands sent from the Raspberry Pi to move the robot, while it also collects and sends the data obtained by the robot connecting to the Raspberry's WAP.

The connection to the *ThingSpeak* server is established through a script that is executed as a sub-process within the main UDP communication routine in the Raspberry, to store the data on the server. The identification and password provided by the platform are used to access the previously created channel. Subsequently, when the data transmission between the WAP and the ESP32 microcontroller is successful, the routine sends the performance parameter to the server to be visualized after a delay of around one minute.

## 3. Results and Discussion

The ANN controller simulation delivered results close to those obtained with the modular robot, with the robot moving through the proposed environments in virtual and real scenarios as shown in Figure 9. In each proposed environment, the modular robot with five modules traveled a distance of 1.8 m. In the ladder environment, the robot surpassed the obstacle. In the L-shaped path, the modular robot turned 45° to the right after advancing 1.05 m. The times and distances obtained in the simulator and the real modular robot are compared in Tables 2 and 3.

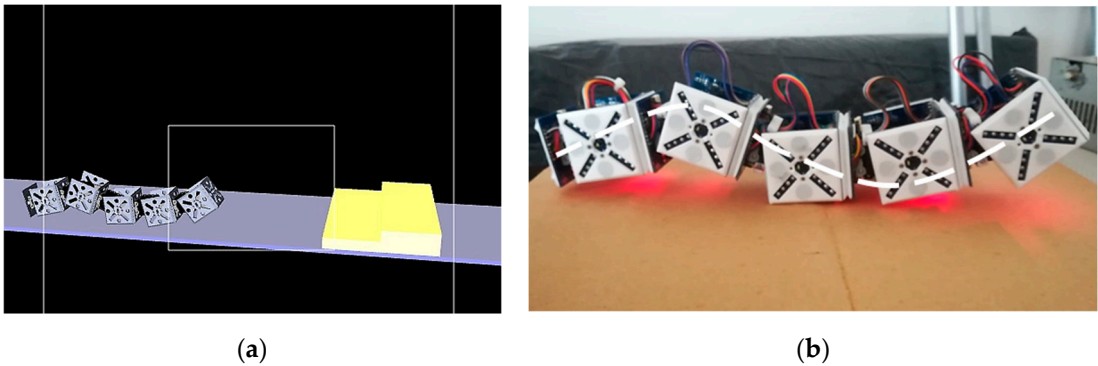

(**a**)                    (**b**)

**Figure 9.** Modular robot movement with five modules: (**a**) Virtual environment; (**b**) Real environment.

**Table 2.** Comparison of the travel times for each environment in the simulator and the real scenario.

| Environment | Time Estimated by the Simulator | Time in Real Scenario | Error |
|---|---|---|---|
| Flat surface | 11 min | 11.6 min | 5.45% |
| Ladder | 13 min | 13.75 min | 5.76% |
| L-shaped | 13 min | 13.7 min | 5.38% |

**Table 3.** Comparison of the traveled distances for each environment in the simulator and the real scenario.

| Environment | Distance Estimated by the Simulator | Distance in Real Scenario | Error |
|---|---|---|---|
| Flat surface | 1.8 m | 1.77 m | 1.66% |
| Ladder | 1.8 m | 1.71 m | 5% |
| L-shaped | 1.8 m | 1.73 m | 3.88% |

Figure 10 shows the margin of error between the training data and the real data during the ANN training, as well as the results of the movement of a module compared to the information stored in the database.

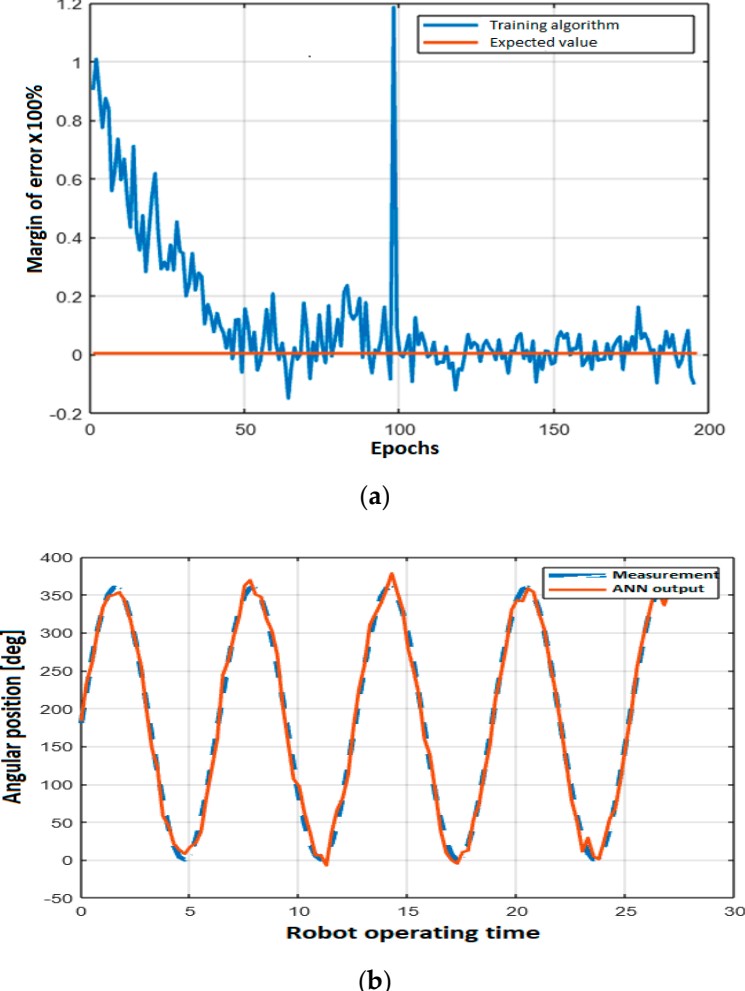

(a)

(b)

**Figure 10.** ANN training: (**a**) Margin of error; (**b**) Movement result vs. measured value, for a single module.

Initially, the most appropriate ANN configuration for robot control was established. Then, the same ANN was tested by changing the number of hidden layers. The results in Figure 11 show that the lowest error was obtained for 25 layers.

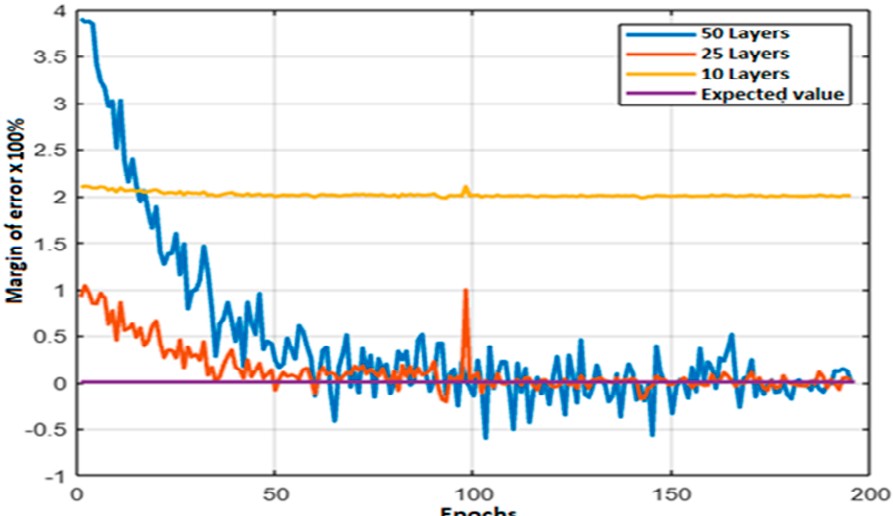

**Figure 11.** ANN margin of error for different numbers of hidden layers.

Finally, the communications network based on SDR was tested. For this, the SNR at different sites was obtained for the Wi-Fi channels connected to the Raspberry Pi, as seen in Table 4, and with a Wi-Fi sensitivity, in the best scenario, of −98 dBm [33]. An adequate SNR value must be above 40 dB, which occurs for most selected channels.

**Table 4.** Average signal-to-noise ratio (SNR) for selected Wi-Fi channels at different sites.

| Place Number | Average SNR (dB) |
| --- | --- |
| 1 | 69 |
| 2 | 46 |
| 3 | 66 |
| 4 | 36 |
| 5 | 58 |
| 6 | 49 |
| 7 | 59 |
| 8 | 53 |
| 9 | 46 |

The Wireshark software was used to find the lost packets and the latency between the Raspberry Pi and the ESP32 module located in the modular robot, for approximately one hour of communication tests. Out of 1647 transmitted and received packets, 0.162% were lost, and the average latency was 12.23 ms.

The average angular velocity information of the modular robot is stored in the cloud through a server hosted by ThingSpeak, as presented in Figure 12.

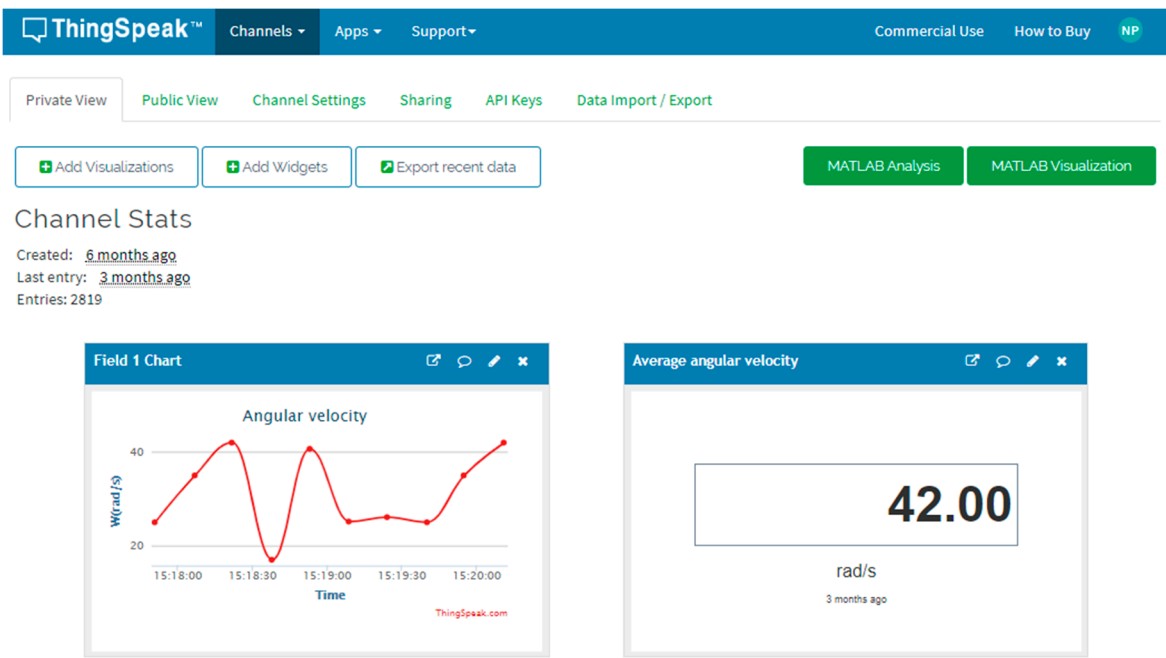

**Figure 12.** Average angular velocity of the modular robot stored in the cloud.

## 4. Conclusions

In this paper, an ANN controller for a modular robot that uses an SDR communication network was presented, where the Wi-Fi channel with the best SNR was selected, and then the information regarding the simulated movements and obtained by the controller was sent to the modular robot, as a contribution to the literature. The distance and time estimated by the simulator did not exceed an error of 6% when compared to those of the real robot, as evidenced in Tables 2 and 3.

The developed ANN controller has 25 layers, two inputs and one output. It predicted the movements of the robot with a training margin of error less than 5%, as seen in Figure 10. Furthermore, this type of strategy is adaptive, which means that a single ANN configuration was required for the robot to move around the environment. Another advantage of this control strategy is that the robot can go through the path even when a module fails, given that the parameters are sent from an external device, such as the Raspberry Pi.

The behavior of the ANN with different configurations of hidden layers showed an optimal operation region for generating the robot controller. This region was found between 20 and 30 hidden layers, since more than 50 layers or fewer than 10 layers caused the training algorithm to not converge satisfactorily. This is depicted in Figure 11. Selecting a number of hidden layers outside the optimal region implies that the controller cannot find an appropriate set of movements for the robot to use to move from one place to another.

The innovative SDR communications network developed transmitted the information corresponding to the robot movements from the simulator to the modular robot, using a WAP developed with the Raspberry Pi and the ESP32 microcontroller located in the robot. This reduced the controller disconnection, and the fault tolerance of the robot was increased. The Raspberry chose, from the establishment of communication, the Wi-Fi channel with the highest SNR, which caused little information loss and low transmission latency compared to in other channels with a lower SNR. Furthermore, the transmitted information was stored and displayed in the cloud, corresponding to the average angular velocity with which the robot moved.

**Author Contributions:** The modular robot and artificial neural network controller were made by H.A.H.; the software-defined radio system was implemented by L.F.P.; the simulator was designed by H.A.H. and executed

by C.A.H.; and H.A.H., L.F.P., and C.A.H. performed the experiments, analyzed the results, drew conclusions, and reviewed and edited the paper. All authors have read and agreed to the published version of the manuscript.

**Funding:** This research was funded by Centro de Investigaciones y Desarrollo Científico (CIDC)—Universidad Distrital Francisco José de Caldas with the project code: 1-273-597-19.

**Acknowledgments:** We express our gratitude to CIDC—Universidad Distrital Francisco José de Caldas for the support. We also appreciate the recommendations of the reviewers.

**Conflicts of Interest:** The authors declare no conflict of interest.

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
