# Peer review of "Artificial Neural Network Controller for a Modular Robot Using a Software Defined Radio Communication System"

_electronics, doi:10.3390/electronics9101626_

Round 1
Reviewer 1 Report
The article is a very pertinent and is focused on a still underdeveloped theme. The authors make a very interesting effort to develop an artificial neural network controller for a modular robot based on an SDR. Thus, a thorough review of the text is recommended to improve English.
the paper examined is very interesting and, overall, shows a high quality standard. In my view, however, some integration and clarification is needed to make it a final product of the highest level. For these reasons I have proposed a "major revision".
Some comments:
- Section 2.1: the description of how the modular robot works is too concise: although research is not focused on its technology, a slightly broader description could be useful for the paper to be a little more "self-standing"
- Section 2.3 “Artificial Neural Network Controller”: please explain better the training algorithm based on the mean squared error method
- Please add some remarks and comment at “Algorithm 1. Control strategy”
- Section 4 “Conclusions”: please better focus the conclusions by highlighting the innovative part
Minor comments:
- Figures 2 a/b are too little: please improve dimension
- Figures 3 a/b are too little: please improve dimension
- Figure 6 is too little: please improve dimension
- Figures 8 a/b/c are too little: please improve dimension
- Figures 10 a/b are too little: please improve dimension
- Figure 11 is too little: please improve dimension
- Due to the huge popularity of the Raspberry system and its application software, Figure 12 does not seem so necessary
I also suggest integrating the References with these papers:
- Jie Zhao, Yanhui Wei, Jizhuang Fan, Jun Shen and Hegao Cai, "New Type Reconfigurable Modular Robot Design and Intelligent Control Method Research," 2006 6th World Congress on Intelligent Control and Automation, Dalian, 2006, pp. 8907-8911, doi: 10.1109/WCICA.2006.1713722.
- Lei Jingtao, Wang Tianmiao and He Yongling, "The modular approach based on functional components division for modular reconfigurable walking robot," 2009 ASME/IFToMM International Conference on Reconfigurable Mechanisms and Robots, London, 2009, pp. 540-544.
Reviewer 2 Report
This is an interesting research paper. There are some suggestions for revision.
1. The motivation is not clear. Please specify the importance of the proposed solution.
2. Please discuss existing solutions.
3. Please highlight your contributions in introduction.
4. Please shorten the discussion of existing work in section 2.1 and 2.2. Discuss the key contents only and add the related references for the remaining ones.
5. As shown in Fig. 7 (a), more details of hidden layers should be discussed. Please explain why the number of hidden layers is 25.
6. More explanations of Tab. 1 should be added. What do FS, La, and L mean in scenarios? Where are the training data from? What is the size of training data?
7. As shown in Algo. 1, please explain where tau=4*10^-8 comes from, how to get the margin of error epsilon, what is the activation function, how to optimize weights.
8. As shown in Algo. 2, it is not clear that movements database and ANN are assigned to theta. Please explain. Where is epochs<1000 from?
9. Please explain where Eq. 2 comes from.
10. It is better to compare the proposed solution with existing solutions.
Round 2
Reviewer 1 Report
Accepted in the present form
Reviewer 2 Report
All my concerns have been addressed. This paper is ready for publication.